# True Parthenogenesis and Female-Biased Sex Ratios in Cicadomorpha and Fulgoromorpha (Hemiptera, Auchenorrhyncha)

**DOI:** 10.3390/insects14100820

**Published:** 2023-10-17

**Authors:** Dora Aguín-Pombo, Valentina G. Kuznetsova

**Affiliations:** 1Faculdade de Ciências da Vida, University of Madeira, 9000-390 Funchal, Portugal; 2Centro de Investigação em Biodiversidade e Recursos Genéticos (CIBIO), 4485-661 Vairão, Portugal; 3Department of Karyosystematics, Zoological Institute, Russian Academy of Sciences, Universitetskaya emb. 1, 199034 St. Petersburg, Russia

**Keywords:** parthenogenesis, thelytoky, female-biased sex ratios, planthoppers, Delphacidae, leafhoppers, Cicadellidae

## Abstract

**Simple Summary:**

The order Hemiptera is incredibly diverse in terms of modes of parthenogenetic reproduction. In this article, we review all currently known data on reproduction by true parthenogenesis, specifically thelytoky, in two major hemipteran suborders, Fulgoromorpha (planthoppers) and Cicadomorpha (leafhoppers). We discuss distribution patterns, ecology, mating behavior, acoustic communication, and the cytogenetic and genetic diversity of parthenoforms. We also highlight examples in which natural populations show a shift in sex ratio toward females and discuss possible causes of this phenomenon, primarily the influence of endosymbiotic bacteria capable of altering the reproductive strategies of the hosts.

**Abstract:**

Insects are renowned for their remarkable diversity of reproductive modes. Among these, the largest non-holometabolous order, Hemiptera, stands out with one of the most diversified arrays of parthenogenesis modes observed among insects. Although there are extensive reviews on reproduction without fertilization in some hemipteran higher taxa, no such analysis has been conducted for the large suborders Fulgoromorpha (planthoppers) and Cicadomorpha (leafhoppers). In both groups, there are species that reproduce by true parthenogenesis, specifically thelytoky, and in Fulgoromorpha, there are species that reproduce by pseudogamy or, more specifically, sperm-dependent parthenogenesis. In this review paper, we give and discuss the only currently known examples of true parthenogenesis in Fulgoromorpha and Cicadomorpha, mainly from the planthopper family Delphacidae and the leafhopper family Cicadellidae. We analyze patterns of distribution, ecology, mating behavior, acoustic communication, and cytogenetic and genetic diversity of parthenoforms and discuss hypotheses about the origin of parthenogenesis in each case. We also highlight examples in which natural populations show a shift in sex ratio toward females and discuss possible causes of this phenomenon, primarily the influence of endosymbiotic bacteria capable of altering the reproductive strategies of the hosts. Our review is mainly based on studies in which the authors have participated.

## 1. Introduction

The origin and evolution of parthenogenesis (parthenos = virgin, genesis = origin), a unique form of reproduction where embryonic development occurs in unfertilized eggs, have puzzled scientists for over a century. Parthenogenesis is a very common, naturally occurring phenomenon among many orders of the animal kingdom, especially invertebrates. The most comprehensive list to date of species reproducing parthenogenetically, as well as a discussion of modes of parthenogenesis in animals, was presented by Bell [1] (see [2,3,4,5,6]). Insects are known to have a wide variety of modes of reproduction and play a central role in our understanding of parthenogenesis, which is often referred to as either unisexual, uniparental, or asexual reproduction. Recently, reviews of known cases of parthenogenesis in different orders of insects have been published [7,8]. In total, parthenogenesis was found in 23 orders, including 2 orders of primary wingless insects, Microcoryphia (=Archaeognatha) and Zygentoma (=Thysanura s. str.), 13 orders of non-holometabolous insects (‘Hemimetabola’), and 8 orders of Holometabola, a monophyletic group that includes most insect species.

Parthenogenesis occurs in various forms and modes, among which the main ones are thelytoky (the mode where females produce only females from unfertilized eggs), arrhenotoky (females produce only males from unfertilized eggs), and deuterotoky (females produce both males and females from unfertilized eggs). Each of these strategies may appear in different groups in different varieties and via an obligatory or facultative manner. Most unisexual insect species or biotypes have arisen as a result of a mutation occurring within a bisexual population, through hybridization events, or by means of polyploidization. The majority of these unisexuals are polyploids, mainly triploids [9,10,11,12,13]. Finally, some endosymbiotic bacteria, most often *Wolbachia* Hertig 1936, can manipulate the reproductive strategy of the host and induce the transition to parthenogenesis [14,15].

The largest non-holometabolous insect order, Hemiptera (bugs), with approximately 97,000–103,590 known species [16,17], has the most diversified modes of unisexual reproduction among insects. The wide variety of reproductive modes and genetic systems of hemipterans make them suitable models for studying how unisexual reproduction emerges and how it is maintained across generations. Hemiptera are taxonomically divided into four monophyletic suborders, including Sternorrhyncha (scale insects, aphids, whiteflies, and psyllids; ~21 extant families), Heteroptera (true bugs sensu stricto; ~54 extant families), Coleorrhyncha (moss bugs or peloridiids; one extant family), Fulgoromorpha (planthoppers; ~20 extant families), and Cicadomorpha (leafhoppers, treehoppers, froghoppers, and cicadas; ~12 extant families) [18,19]. Existing studies show that parthenogenesis is unevenly distributed among and within these phylogenetic lineages. Moreover, trends in the evolution of unisexual forms exhibit significant disparities across different suborders [7,20].

The suborder Sternorrhyncha has the most notable instances and types of unisexual reproduction. This is especially true of scale insects (Coccoidea), in which bisexual reproduction is often combined with numerous aberrant modes of reproduction [21,22,23,24,25,26,27]. It is worth mentioning various modes of reproduction observed in scale insects, including haplodiploidy, diploid arrhenotoky with Paternal Genome Elimination (PGE; males are haploid) or Paternal Genome Heterochromatinization (PGH; males are diploid) during embryogenesis, automictic and apomictic thelytoky (both obligate or facultative), deuterotoky, and hermaphroditism, which is in fact extremely rare in insects (Table 1). Most aphids (Aphidoidea) typically reproduce by cyclical parthenogenesis, alternating one annual (sometimes biannual) bisexual generation with several (or numerous) unisexual (all-female) generations reproducing by apomictic parthenogenesis. The bisexual generation may be lost secondarily, so that reproduction is then exclusively by thelytoky [28,29,30,31,32,33,34]. Although very little is known about the genetic makeup of whiteflies (Aleyrodoidea), it has been shown in some model species (e.g., *Bemisia tabaci* (Gennadius, 1889) and *Aleurodicus rugioperculatus* Martin, 2004) that the females are derived from fertilized eggs, whereas males are parthenogenetically produced by arrhenotoky [35,36]. In this system, males inherit their mother’s but not their father’s genes. Psyllids (Psylloidea) are almost exclusively bisexual. However, some species are known to reproduce by parthenogenesis, at least in separate populations [37,38], and the issue of parthenogenesis in these species, its origin, and its evolutionary role have been the subject of many articles in recent years (e.g., [39,40,41,42]). There are no known cases of unisexual reproduction in peloridiids (Coleorrhyncha), and only a few thelytokous (most likely facultatively thelytokous) species have been reported in true bugs (Heteroptera) [43,44].

The last two suborders, Fulgoromorpha (also known as planthoppers) and Cicadomorpha (also known as leafhoppers), constitute the focal point of this present paper. Within these suborders, documented instances of parthenogenesis are relatively scarce, numbering fewer than a dozen documented cases of parthenogenesis [7]. Notably, the reproductive patterns of some species within these suborders have been studied in great detail. In both suborders, there are known species reproducing by true parthenogenesis (specifically, thelytoky), and in Fulgoromorpha, there are known species reproducing by pseudogamy or, more specifically, sperm-dependent parthenogenesis [45,46,47,48,49,50,51]. It is important to note that planthoppers are the only group of Hemiptera demonstrating such a rare mode of reproduction, which we assume to discuss later in a special publication. Here, we provide an extensive, focused review of the occurrence of true parthenogenesis in different families of Fulgoromorpha (mainly Delphacidae) and Cicadomorpha (mainly Cicadellidae). In addition, we discuss cases where assumptions of parthenogenesis in a species or population have been made solely on the basis of the rarity or complete absence of males in collections from natural populations, even though their exact reproductive strategy has not been confirmed in any way. We focus on cases in which natural populations show a highly female-biased sex ratio and discuss potential causative factors for this phenomenon, with particular emphasis on the influence exerted by endosymbiotic bacteria capable of altering the reproductive strategies of their hosts. Our review is mainly based on studies in which we—the authors—have participated.

## 2. A Brief History of Parthenogenesis in Fulgoromorpha and Cicadomorpha

The suborders Fulgoromorpha and Cicadomorpha, previously grouped in the suborder Auchenorrhyncha, are two worldwide specious groups with more than 43,000 valid species [52,53] distributed roughly in 32 families. Fulgoromorpha contain one superfamily, Fulgoroidea, with 20 recognized families, and Cicadomorpha include the superfamilies Cicadoidea (cicadas, 2 families), Cercopoidea (spittlebugs, 5 families recognized here, but 3 in Hamilton [54]), and Membracoidea (leafhoppers and treehoppers, 5 families). 

The history of parthenogenesis in Auchenorrhyncha began with experimental studies on the American species of leafhoppers, *Agallia quadripunctata* (Provancher, 1872) (Cicadellidae, Agallinae, Agalliini), in which different populations consist predominantly of females ([55] and references therein). Since the mid-1970s, a group of scientists from the Agricultural University of Wageningen (The Netherlands), including S. Drosopoulos, C.J.H. Booij, A.J. De Winter, P.W.F. De Vriejer, and C.F.M. Den Bieman, has made significant contributions to the study of parthenogenesis in Auchenorrhyncha. Drosopoulos [45,46] followed by Den Bieman and Eggers-Schumacher [47] and Den Bieman [48,49,50] showed that planthoppers of the genera *Muellerianella* Wagner, 1963, and *Ribautodelphax* Wagner, 1963 (both from the family Delphacidae, Delphacinae), respectively, reproduce by pseudogamy. Den Bieman and de Vrijer [56] were the first to describe a case of true parthenogenesis in the planthopper genus *Delphacodes* Fieber, 1866 (Delphacidae, Delphacinae, Delphacini). This group of researchers continued to work intensively over the next 15 years with a multidisciplinary approach that involved studies on morphology, cytogenetics, population genetics, hybridization, mating acoustic communication, ecology, and distribution. More recently, true parthenogenesis has been documented for the leafhopper genus *Empoasca* Walsh, 1862 (Cicadellidae, Typhlocybinae, Empoascini) from Madeira Island, where three apomictic morphotypes have been discovered and extensively studied [57,58]. A recent study [59] suggested that the sex-distorting bacteria of the genus *Rickettsia* could be responsible for the origin of parthenogenesis in these leafhoppers. There are other published records of bisexually reproducing Auchenorrhyncha species. In these cases, vertically transmitted endosymbiotic bacteria belonging to the genera *Wolbachia*, *Cardinium* Zchori-Fein et al. 2004, *Arsenophonus* Gherna et al. 1991, *Spiroplasma* Saglio et al. 1973, and *Rickettsia* da Rocha-Lima, 1916—collectively referred to as reproductive parasites—induce a shift in the sex ratio towards females. These infections compel infected females to produce offspring daughters in the absence of mating (e.g., [60,61]). In addition, in some species, parthenogenesis is suspected only because there is a bias towards females or a complete absence of males in the field collections (e.g., [62]). 

## 3. A Brief Overview of the Patterns and Origins of True Parthenogenesis 

Strictly speaking, parthenogenesis refers to reproduction by virgin females. That is, females give rise exclusively to females without the need for mating. This type of parthenogenesis, known as true parthenogenesis, can manifest in two forms: obligatory parthenogenesis and facultative parthenogenesis (Table 1). In the first case, species reproduce solely by parthenogenesis, while in the second case, parthenogenesis is a sporadic means of reproduction. A particular case of facultative parthenogenesis, termed cyclical parthenogenesis, is when the sexual and asexual modes of reproduction alternate according to environmental conditions [6]. This mode of reproduction is very common in Sternorrhyncha, specifically in aphids, while in Cicadomorpha and Fulgoromorpha, obligate parthenogenesis is the only type so far reported. Depending on whether new parthenogenetic forms arise from unfertilized eggs that have undergone or not meiosis, true parthenogenesis can be further divided into automictic and apomictic, respectively. Furthermore, based on the sex produced by parthenogenesis, a distinction is made between arrhenotoky (producing only males), thelytoky (producing only females), and deuterotoky or amphitoky, where eggs develop both into males and females [6]. In thelytoky, offspring can be diploid, triploid, or polyploid, though triploidy is the more common form. In most cases of thelytokous parthenogenesis, eggs do not suffer chromosome reduction through meiosis (apomixis); nonetheless, in some groups, meiotic reduction of genetic material is followed by a subsequent restoration of diploidy in offspring (automixis) [63]. 

There are three primary origins of thelytoky [10,64]. The first of these is associated with hybridization, i.e., crosses between two bisexual species resulting in the generation of parthenogenetic hybrids. The second origin has to do with genetic mutations, i.e., spontaneous loss of “sexual” reproduction due to mutations in genes responsible for the production of bisexual forms and the successful occurrence of meiosis. The third origin is associated with bacterial endosymbiosis, specifically infection by inherited bacteria capable of invading the host species and sustaining their presence by manipulating the reproduction of infected hosts. The latter mode, involving bacterial endosymbiosis, represents the most extensively studied cause of parthenogenesis in insects [65]. In the last decades, due to advances in microbiology, it has been discovered that symbionts of the *Flavobacterium* clade, such as *Wolbachia*, *Cardinium*, *Arsenophonus*, and *Spiroplasma*, and the *Rickettsia* clade, can be reproductive parasites [66]. Notably, in the majority of cases, it is *Wolbachia* that has been suggested to infect over 70% of arthropod species [67], including more than a million insect species [14]. Bacterial endosymbionts live inside the cytoplasm in reproductive tissues. To increase the proportion of infected females, they can modify the reproductive biology of the host, skewing the sex ratio towards females. In different insects, host reproduction can be affected by the induction of parthenogenesis, feminization of males, male killing, and the induction of cytoplasmic incompatibility between gametes (CI), a form of embryonic lethality in crosses between males and females with different infection status [15]. In Auchenorrhyncha, the effects of these bacteria on reproductive biology have been studied in only a few species; however, they have been shown to cause phenomena such as CI, male killing, and feminization of males [60,61,68,69,70,71,72]. A recent study [59] suggested that *Rickettsia* might induce parthenogenesis in the leafhopper genus *Empoasca*.

## 4. Well-Documented Cases of True Parthenogenesis in Fulgoromorpha and Cicadomorpha

Reproduction by true parthenogenesis has been documented in two genera of the leafhopper family Cicadellidae, specifically in *Agallia* Curtis, 1833, and Empoasca, as well as in the planthopper family Delphacidae, specifically in the genus *Delphacodes*. To study the reproduction of these species, a wide range of methods and approaches were used, including hybridization experiments, analysis of morphology, cytology, biology, ecology, host plant relationships, distribution, and mating acoustic behavior. 

### 4.1. The Leafhopper Family Cicadellidae


**The species**
*Agallia quadripunctata*
**(Four-spotted Clover Leafhopper)**


The true representatives of the genus *Agallia* are likely to occur only in the Old World, whereas the approximately 110 New World species currently placed in the genus possibly belong to one or more yet undescribed genera ([73] and references therein). Several authors have independently reported that males of *Agallia quadripunctata* (Figure 1a) are rarely found in collected material ([55] and references therein), so the question has arisen as to what mode of reproduction this species has. To answer this question, Black and Oman [55] collected nymphs of *A. quadripunctata* around Washington, DC, and set up rearing experiments on plants of crimson clover, *Trifolium incarnatum* L., 1753 (Fabales, Fabaceae). The resulting adults successfully reproduced over a two-year period and produced no males. From this offspring, 30 nymphs were individually caged, and again, the progeny consisted only of females. The authors concluded that *A. quadripunctata* reproduces parthenogenetically, at least in the northeast of the United States. 

Distribution and Ecology of Parthenoforms

*Agallia quadripunctata* occurs throughout the Nearctic region, from southern Canada to Mexico and Cuba [74,75,76,77]. It appears to be common in the eastern USA, although records are still lacking for a significant part of the country (Table 2). Males of *A. quadripunctata* in the American Museum of Natural History (unpublished data) and some published records (i.e., [78]) suggest that bisexual populations of this species occur in the USA at the eastern and western limits of its range, in New York, Connecticut, Illinois, and Oregon. Unisexual forms are found in the northernmost parts of the USA, possibly coexisting with bisexual populations in some areas, and appear to be spreading into southern Canada (e.g., Mova Scotia) [77,79,80]. These preliminary data suggest that apomictic forms may be distributed further north than their bisexual relatives. Parthenogens are most abundant in marginal areas, such as deserts, islands, high altitudes and latitudes, and areas formerly covered by ice or exposed to extreme aridity (reviewed by Kearney [11]), a pattern known as geographical parthenogenesis, observed in species with mixed reproductive strategies [81,82]. This elevational pattern is a common phenomenon in many species with unisexual reproduction, a trend shared by Hemiptera [44]. Another prediction is that unisexual forms are more common in disturbed environments. Although *A. quadripunctata* is common in anthropogenic environments, such as burned grasslands [83], and in agricultural areas where it may even become a pest [77], it is not yet known whether populations in these habitats are unisexual, bisexual, or both. *A. quadripunctata* is polyphagous and lives among the roots of grasses, sedges, and other plants in a wide range of moist habitats, such as wet/mesic grasslands [83,84], as well as in shaded and open woodlands [85,86]. This polyphagous species [87] has as its main hosts many species of Fabaceae [55,80,88,89] and some species of Amaranthaceae [90] and Ulmaceae [91]. Whether this wide ecological range is related to its unisexual mode of reproduction is not yet known.

Origin and Genetic Diversity: Rare Males

The presence of males in *A. quadripunctata* suggests that this species probably reproduces both bisexually and by parthenogenesis. However, the lack of information on the ploidy level of females, the spermatogenesis of rare males, and their role in populations makes it impossible to conclude whether parthenogenesis in this species is facultative or obligate. Indeed, Black and Oman [55] suggest that “the presence of rare males in some populations of *A. quadripunctata* may indicate that at times the species does reproduce bisexually”. On the other hand, such males may be non-functional. Rare non-functional (spanandric) males occur in other hemipterans, e.g., in the triploid parthenogenetic psyllid species *Cacopsylla myrtilli* (Wagner, 1947) (Psylloidea) [92]. However, there is no information available to draw any conclusions as to whether parthenogenesis in *A. quadripunctata* could be due to interspecific hybridization or whether some other mechanism is involved. In the United States, the *quadripunctata* group [93] includes four closely related bisexual species with which interspecific crosses are theoretically possible, two of which (*Agallia pumila* Oman, 1971 and *A. excavata* Oman, 1933) occur in the western USA and two others (*Agallia deleta* Van Duzee, 1909 and *A. obesa* Oman, 1933) in the eastern USA [93]. Crosses of *A. quadripunctata* with the closest species, *A. pumila*, seem unlikely because, first, *A. quadripunctata* is very rare in the western USA and, second, the unisexual forms are found just in the opposite part of the country. On the other hand, *A. pumilia* has been described as a species distinct from *A. quadripunctata* only because it is smaller and reproduces bisexually [93] and, therefore, it may, in fact, be a bisexual form of *A. quadripunctata*. The other two species, *A. deleta* and *A. obesa*, from the southeastern USA, appear to have contact zones with *A. quadripunctata*, raising the possibility of interspecific encounters with it.


**The genus**
*Empoasca*


This cosmopolitan genus, which includes more than 880 described valid species [94,95], provides the most remarkable cases of true parthenogenesis not only within Cicadellidae, but also for Auchenorrhyncha as a whole. In the 2000s, a noteworthy discovery was made, wherein three distinct all-female populations (referred to as parthenoforms) with distinctive morphological and cytogenetical characteristics were identified. These populations attributed to the genus *Empoasca* (Figure 1b) were found on the small island of Madeira, located off the western coast of Africa [57]. Rearing experiments proved that these parthenoforms, provisionally named morphotypes A, B, and C, reproduced in the absence of males for either 10 generations (A and C) or two generations (B). Morphotypes A, B, and C appeared to have different chromosome numbers, 31, 27, and 24, respectively. Simultaneously, males of the local bisexual species, *Empoasca alsiosa* Ribaut, 1933, and *E. fabalis* DeLong, 1930, were shown to have 2n = 17(16 + X) and 2n = 21(20 + X), respectively (Table 2). The study revealed that the reproduction of parthenoforms is of the apomictic type [57]. It has recently been shown that morphotypes differ in wing venation both from each other [58] and from the aforementioned bisexual species. 

In Europe, the genus *Empoasca* contains many pests of crops and, therefore, is widely studied (e.g., [96,97,98,99]); in spite of this, other cases of asexual reproduction have not been known. It is worth mentioning, however, that Parh [100], during an investigation of the *Empoasca* complex associated with cowpea crops Vigna unguiculate (Linnaeus, 1759), Van Eselt (Fabales, Fabaceae), in Nigeria, observed the presence of “less harmful unidentified parthenogenetic females” in addition to the main pest species, *Empoasca dolichi* Paoli, 1930, a. Akingbohungbe [101] speculated that it might have been *Empoasca confusania* Ghauri, 1979, because its terminalia matched those of females of this species and, to a lesser extent, the terminalia of *E. dolichi*. Although both *E. dolichi* and *E. confusania* coexist on cowpea with the parthenogenetic form in southeastern Nigeria, rearing experiments proved that they both reproduced sexually [102]. More recently, the first author of this paper discovered an all-female population similar to *E. dolichi* but distinct from the above-mentioned Madeira morphotypes during fieldwork on the Cape Verde islands of Santo Antão and Santiago, suggesting the possibility of several parthenogenic lines in the genus *Empoasca*. 

Distribution and Ecology of Parthenoforms

The all-female populations of *Empoasca* are all linked to disturbed moist habitats. They are polyphagous, associated mainly with agricultural, ruderal, and ornamental plants, and are common in anthropogenic environments, particularly at low altitudes. They often inhabit the same habitats, and some even coexist on the same plants with the three bisexual species, *Empoasca fabalis*, *E. alsiosa*, and/or *Asymmetrasca decedens* (Paoli, 1932). Likewise, the aforementioned unidentified all-female *Empoasca* from Nigeria is also polyphagous, inhabiting both moist habitats and agricultural areas. This species also shares the same host plants, such as the wild bean of the genus *Phaseolus* Linnaeus, 1753 (Fabales, Fabaceae), with the parthenoforms found in Madeira. These polyphagous all-female populations occur in the rainforest and the Guinea savannah ecoregion, where cowpea, *Vigna unguiculata*, is very abundant [103]. It is worth noting that four of the five currently known parthenogenetic forms of leafhoppers inhabit islands (Madeira and possibly the Cape Verde Archipelago). Although further research is needed to confirm whether this island pattern exists, the present data on *Empoasca* fit well with predictions of geographical parthenogenesis. It is important to note that studies on parthenogenesis within island ecosystems remain relatively limited. However, recent observations of all-female populations of species such as *Ischnura hastata* (Say, 1839) (Odonata, Coenagrionidae) and *Nephus voeltzkowi* (Weise, 1910) (Coleoptera, Coccinellidae) within the nearby Azores Archipelago, where parthenogenesis was previously undocumented, provide additional support to this idea [104,105]. In the case of the above-mentioned parthenogenetic form in Nigeria, while its distribution remains understudied, available data suggest a potentially widespread occurrence, especially within agricultural habitats in the south of the country where it occurs [101,102]. Consequently, the association of this form with artificial habitats may also be consistent with the prediction of geographical parthenogenesis.

Origin and Genetic Diversity: Clones, Rare Males

Some indirect evidence suggests that the parthenogenetic forms of *Empoasca* leafhoppers on Madeira did not come from nearby continental areas but probably originated within this island [57]. The genetic relationship of the parthenoforms with each other and with the closest bisexual species remains unclear. However, some preliminary hypotheses based on chromosomal and morphological data have been proposed for all-female populations on Madeira [57]. Chromosomal analysis showed that the parthenoform B (27 chromosomes) is triploid with 2n = 3x = 24 + XXX, as evidenced, in particular, by the presence of three easily distinguishable large chromosomes in its karyotype. If this form originated on Madeira, it can be speculated that it is autopolyploid and arose by the triplication of the haploid set of *E. alsiosa* (n = 8 + X). Autotriploidy could result from the fertilization of an unreduced diploid gamete or the fertilization of a haploid egg by more than one sperm (polyspermy). Examples of autotriploidization are widespread in insects [3]. As for the presumably triploid parthenoform A (31 chromosomes), it could have arisen because of hybridization between diploid bisexual species with different numbers of chromosomes. On the island of Madeira, bisexual species have different origins, with one species from the Americas (*Empoasca fabalis*), one from the Palaearctic region (*Asymmetrasca decedens*), and a third one from the eastern Mediterranean (*Empoasca alsiosa*). It is not surprising that after their introduction to Madeira, these species established themselves in similar ecologically disturbed habitats, and the frequent interactions between them on this relatively small island facilitated interspecific mating and the appearance of hybrid forms. 

Parthenoforms are expected to be more common on islands than on the mainland. This is because, firstly, parthenoforms are theoretically better colonizers since one female is sufficient to establish a new population, and, secondly, if they appear on islands, it is potentially easier for them to settle there due to lower competition and more accessible, poorly explored habitats. Allopolyploidy is a common phenomenon in animals and is often accompanied by repeated hybridization and backcrossing of hybrid offspring. The biological and ecological requirements of bisexual *Empoasca* species from Madeira do not differ significantly, to the extent that their interactions are constrained. In fact, unisexual lineages often exhibit a wide range of genetic backgrounds, characterized by the presence of multiple coexisting clones [106].

As for the parthenoform C (24 chromosomes), the karyotype analysis, primarily focusing on chromosome size, did not provide conclusive evidence to determine whether it is diploid (2n = 22 + XX) or triploid (2n = 3x = 21 + XXX). Furthermore, it is noteworthy that rare males, accounting for less than 0.05% of the population, have been found. Some of these males exhibit morphological resemblances to females belonging to Morphotype C. Interestingly, their genital structures resembled more those of a bisexual species not previously recorded on Madeira—the African *Empoasca distinguenda* Paoli, 1932. In some parthenogenetic species, rare males, if present, can mate with “sexual” females and spread asexuality-related genes between bisexual populations [107,108]. When rare males retain the ability to mate with females of related bisexual lineages, they can pass the genes conferring parthenogenesis to their offspring [82,107,109], a mechanism termed ”contagious parthenogenesis” [10].

Another possible origin of parthenogenesis in *Empoasca* could be infectious. To find out if these parthenogenetic morphotypes have bacteria that cause sexual disorders, the presence of *Wolbachia*, *Cardinium*, *Rickettsia*, and *Arsenophonus* was diagnosed both in them and in three bisexual relatives of *Empoasca* [59]. *Wolbachia* was found in both parthenogens and bisexual species, whereas Rickettsia was present only in parthenoforms, suggesting that asexual reproduction, at least in some of them, may be of infectious origin. Further studies may resolve this mystery. As for all-female *Empoasca* from southern Nigeria, the finding on cowpea of specimens morphologically intermediate between *E. dolichi* and *E. confusania* allows us to suggest that they may be the result of hybridization between these two species [101]. This assumption finds further support in the morphological observations of male genital structures, which revealed a combination of characteristics resembling those found in both *E. dolichi* and *E. confusania*. This observation suggests that interspecific crosses may be a common phenomenon in situations where both species coexist on cowpea [101,110].

### 4.2. The Planthopper Family Delphacidae


**The genus**
*Delphacodes*


Females of the species of the genus *Delphacodes* are morphologically indistinguishable. In northwestern Greece (Prassino, ~1000 m, Florina), in a wet biotope on *Carex riparia* Curtis, 1783 (Poales, Cyperaceae), an all-female population was found, morphologically similar to the *Delphacodes* species [111]. No males were ever found in this locality despite intensive sampling in different seasons [56,111]. To see how these females reproduce, adults and larvae were collected from the host plant and reared in the laboratory [56]. For more than ten generations, only females were obtained, providing clear evidence of reproduction by true parthenogenesis. This represents the first and, to date, the sole known instance within the planthoppers (Figure 2a). The karyotype analysis of the laboratory-reared females consisted of 44 univalents. According to Den Bieman and De Vrijer [56], these females were triploid, characterized by “two female genomes (28 + 2X) and one male genome (14 + 0)”. Insemination of triploid females by *Delphacodes capnodes* (Scott, 1870) males resulted in all triploid females but did not increase ploidy [56]. This observation suggests that matings with bisexuals do not give rise to new clones.

Distribution and Ecology of Parthenoforms

Parthenogenetic populations of *Delphacodes* are currently known only from northern Greece, where they occur at high altitudes (>1000 m) (Table 2). Since the Greek planthopper fauna and its food plants are well studied, it is possible to assume that these all-female populations occur in the mountains [111]. However, it should be noted that triploid females do not have reliable diagnostic characters [112] and therefore can easily be overlooked. In Europe, the genus *Delphacodes* is represented by ten bisexual species [113,114,115], four of which occur in Greece: *D. capnodes*, *D. nastasi* Asche & Remane, 1983, *D. venosus* (Germar, 1830), and *D. schinias* Asche & Remane, 1983 [111]. None of these bisexual species were found in the same geographic area as the all-female populations under study. If the all-female Greek populations originated from one of these species, it would be reasonable to expect other clones in different regions of Europe. Indeed, there have been reports of all-female populations of *Delphacodes* in the Czech Republic [116] and Germany [113,114]. However, there remain doubts regarding the exclusively female composition of these populations [116]. These doubts arise primarily due to the limited number of individuals in the samples and the collection method used, specifically pitfall traps, which are known to capture more females than males. Regardless of the distribution, the presence of parthenoforms at high altitudes is consistent with the general tendency of apomictic forms to be at higher altitudes compared to their biparental relatives. Nothing is known about the ecology of parthenogenetic populations of *Delphacodes* in their natural habitat (Ibid.), except that they live in a wet biotope on *Carex riparia*. However, we can assume that they may share some ecological similarities with *D. capnodes*. This bisexual relative lives close to the ground in *Sphagnum* hummocks in tall-sedge swamps, reeds, and intermediate bogs associated with *Carex* spp. (*Carex acutiformis* Ehrh., 1789, *C. acuta* L., 1753, and *C. riparia*) and *Eriophorum angustifolium* Honck., 1782 (Poales, Cyperaceae) [114]. Most species of this genus are host-specific, with host plants serving as a substrate for acoustic signaling and communication between the sexes. Consequently, differences in host plant specificity can potentially function as an effective mechanism of acoustic isolation in substrate communication [117].

Origin and Genetic Diversity: Mating Behavior

Some work has been performed to clarify the origin of the triploid parthenogenetic population from northwestern Greece, i.e., whether it originated from intraspecific whole genome duplication (autopolyploidy) or from interspecific hybridization. In northwestern Greece, the species *Delphacodes capnodes*, *D. nastasi*, and *D. venosus* occur [111], so hybrids between these species are theoretically possible. However, the biology of parthenogenetic populations is unknown, and there are no data on whether biological and/or ecological differences may limit the coexistence of unisexuals and bisexuals. If parental species such as *D. capnodes* have only one generation per year [115,118], temporary isolation may occur. Although the all-female population could not be accurately associated with any of the four Greek bisexual species, acoustic mate signals help to clarify the identity of closely related taxa [119]. In planthoppers, mating choice and mating behavior are mediated by the exchange of species-specific substrate-borne vibration signals (Figure 3) [119,120]. Therefore, females of one species are not expected to respond to the calling songs of another species. Den Bieman and De Vrijer [56] studied the mating behavior of parthenogenetic females from the Greek Prassino population and compared it with that of a bisexual population of *D. capnodes* from *Eriophorum angustifolium* in the Netherlands. The authors conducted four separate trials, in which they caged 10 males with 10 females from unisexual populations. Interestingly, the behavior, structure, and length of the calling songs emitted by both bisexual and parthenogenetic females were similar. Furthermore, both groups responded strongly to the calling songs produced by the males (Figure 3). Before mating, they exchanged only a few vibrational signals, after which they willingly engaged in copulation for four to ten minutes. All triploid females were successfully inseminated, and their offspring were again all-female and triploid with 44 univalents (n = 40 females). The successful insemination of all females in this experiment is remarkable because mating with males of bisexual allopatric congeneric species is usually much less successful. The mating success of parthenogenetic females and the similarity of their mating behavior to *D. capnodes* females suggest that the parthenogenetic lineage might have arisen from a *D. capnodes*-like ancestor. Claridge and De Vrijer [119] hypothesized that because parthenogenetic females behaved in many respects like the pseudogamous forms of the planthoppers *Muellerianella* and *Ribautodelphax*, parthenogenesis in *Delphacodes* may have evolved through an ancestral pseudogamous stage and subsequently evolved to be free of obligate coexistence with biparental species.

## 5. Suspected Cases of Parthenogenesis in Fulgoromorpha and Cicadomorpha Based on Faunistic Studies

Parthenogenesis within a species or population can be proved either by chromosome analysis or by experimental crossbreeding in the laboratory. However, there are examples where the inference of parthenogenesis is drawn from field observations, specifically based on the predominance of females or the complete absence of males. We have already mentioned such examples above, and in this context, we will add some more examples from the leafhopper family Cicadellidae and the planthopper families Delphacidae and Issidae.


**Cicadellidae**


The unusual female-biased sex ratios reported in faunistic studies indicate that parthenogenesis may occur in *Ledromorpha planirostris* (Donovan, 1805), *Laburrus amazon* Emeljanov, 1962, and *Anaceratagallia kerzhneri* Emeljanov, 1987 (Table 3). *Laburrus amazon* (Figure 2b) and its host plant, *Artemisia pauciflora* Weber ex Stechm., 1775 (Asterales, Compositae), are widely distributed from the lower Volga to Altai, encompassing a strip of semi-deserts and northern desert regions [122,123]. The only two males found so far (lower Volga) have poorly developed genitalia. This species is common, abundant, and easy to identify; therefore, the scarcity of males suggests that it may reproduce by parthenogenesis. *Anaceratagallia kerzhneri* (Figure 4b) is also widespread throughout the stepped zone, from Mongolia to the Maritime Territory of eastern Russia. Only one male was collected in eastern Russia (Amur region), which suggests that it may reproduce by parthenogenesis in Mongolia and Primorsky Krai. In Europe, several all-female populations of *Arboridia* Zachvatkin, 1986, have been found, suggesting reproduction by parthenogenesis. In this case, as the females of this genus have no diagnostic characteristics, it is not possible to speculate to which bisexual species they may be related. Especially remarkable is the suspicion that the largest leafhopper species, *Ledromorpha planirostris*, may reproduce parthenogenetically throughout its whole range (Figure 4a) [124,125,126]. This species feeds on *Eucalyptus* spp. [127] and is endemic to Australia and fairly common in coastal areas of the eastern region, from northern Queensland to South Australia (Adelaide) and Tasmania [126,127,128]. Although dozens of female specimens have been collected, no males are known [127]. The single male reported as the “holotype” that was illustrated by Donovan in 1805 [129] is most likely a female (see [127]). There is also a suspicion of reproduction by parthenogenesis in its closest relative [130], *Ledracostis gunnensis* Evans, 1937. 


**Issidae**


A case of all-female populations in xeric areas was found in Issidae. This family comprises nearly 1000 species distributed worldwide [131]. Populations of *Scorlupella montana* (Becker, 1865) have unusually female-biased sex ratios [62] (Table 3). This Palaearctic species primarily inhabits the steppe region with xerothermophile vegetation, often occurring at high altitudes in fields and sandy areas [132,133,134]. In Iran, it was collected on *Astragallus* sp. within steppe highlands in proximity to agricultural areas and grasslands, ranging from 1330 to 2600 m [131,135]. In Kyrgyzstan, it has been collected in arid regions, mountain steppes, semi-desert environments, and Picea forests [133]. *Scorlupella montana* is widely distributed from Europe to Central Asia [122]. Despite its extensive geographical range and large number of specimens available, only two males are known from Turkey [131] and the northwestern Caucasus region [62].

Interestingly, all mentioned examples of putative parthenogenesis refer to widely distributed species, mainly from dry habitats, deserts, or semidesert regions, with a narrow host plant range. Many cases of geographical parthenogenesis have been described from desertic areas, and some examples of these have been extensively studied in Australia (e.g., [11,135]). Although such patterns of geographic parthenogenesis are common in many groups of organisms, they are not universal. Many cases of parthenogenesis do not necessarily follow any specific ecological or distributional pattern [136]. 

## 6. Conclusions

Reliably confirmed cases of true parthenogenesis in Auchenorrhyncha are few and limited to members of only three genera in the planthopper family Delphacidae (*Delphacodes*) and the leafhopper family Cicadellidae (*Agallia* and *Empoasca*). However, data from faunistic studies, cytogenetics, and crossbreeding experiments suggest that true parthenogenesis may be more common in both Fulgoromorpha and Cicadomorpha than previously thought. We have also shown that, in some cases, the transition to unisexuality can be induced by endosymbiotic bacteria. Further research could focus on populations in which the sex ratio is strongly skewed towards females, as well as on identifying the mechanisms underlying parthenogenesis in planthoppers and leafhoppers. As for the leafhopper genus *Empoasca*, to which our interests are addressed and our studies are devoted, new approaches are urgently necessary. First, for studies at the molecular-genetic level, the detection of several cases of parthenogenesis in one genus at once, while parthenogenesis in Auchenorrhyncha in general is extremely rare, testifies, apparently, to the lability of the sexual reproductive system in these leafhoppers. This also suggests that the genetic basis for the transition to parthenogenesis in *Empoasca* species likely involves a small number of genes. Therefore, further studies may be directed at identifying individual genes responsible for this reproductive transition and determining the mechanisms by which these genes cause such transitions. Our studies of the genus *Empoasca* have shown that *Rickettsia* bacteria are exclusively present in unisexual populations, suggesting a connection between infection and parthenogenetic reproduction. Further research is needed, including searching and studying other parthenoforms of *Empoasca*, to prove such a relationship. Finally, it is important to understand whether parthenogenetic populations of *Empoasca* on Madeira confirm a prediction of the hypothesis of geographic parthenogenesis that unisexual reproduction tends to occur on islands. 

## Figures and Tables

**Figure 1 insects-14-00820-f001:**
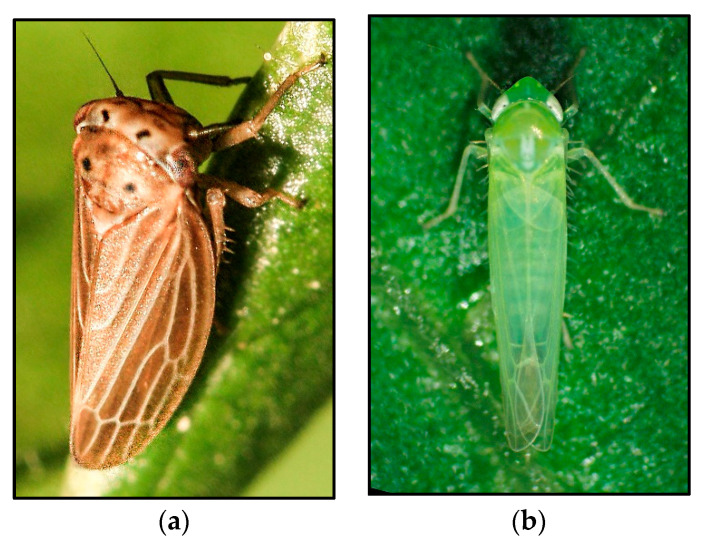
Proven cases of parthenogenesis in Cicadellidae. (**a**) Female of *Agallia quadripunctata*, photo credit: Dan Leeder; (**b**) Morphotype A of *Empoasca* from Madeira, photo credit: Nelio Freitas.

**Figure 2 insects-14-00820-f002:**
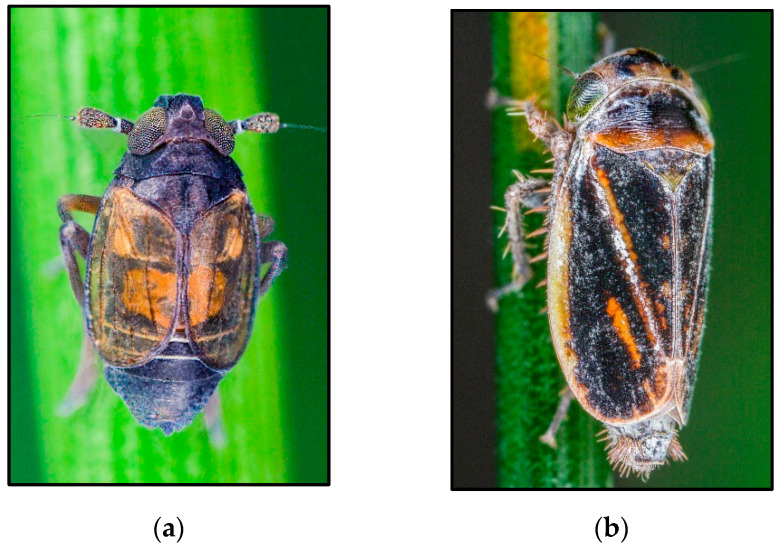
Examples from Delphacidae and Cicadellidae with proven or suspected parthenogenesis. (**a**) Delphacidae: *Delphacodes venosus*; (**b**) Cicadellidae: *Laburrus quadratus* (Forel, 1864). Photo credit: Gernot Kunz.

**Figure 3 insects-14-00820-f003:**
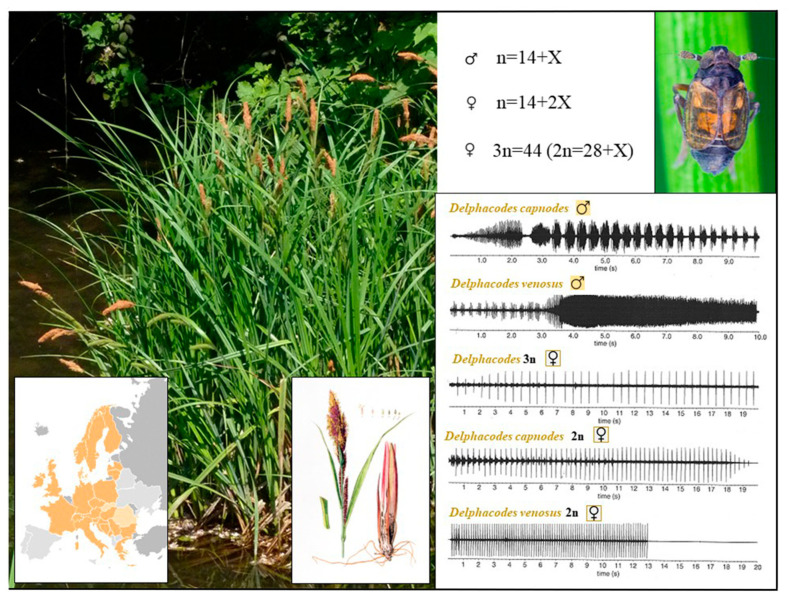
Female calls of triploid parthenogenetic *Delphacodes* and male and female calls of closely related bisexual diploid species, *D. capnodes* and *D. venosus* (Delphacidae). Oscillograms redrawn from records by C.F.M. Den Bieman and P.W.F. de Vrijer published in [121]. Photo credits: *Delphacodes venosus* by Gernot Kunz; illustration of *Carex riparia* by Christiaan Sepp (public domain); photo of *C. riparia* by Bertrant Bui.

**Figure 4 insects-14-00820-f004:**
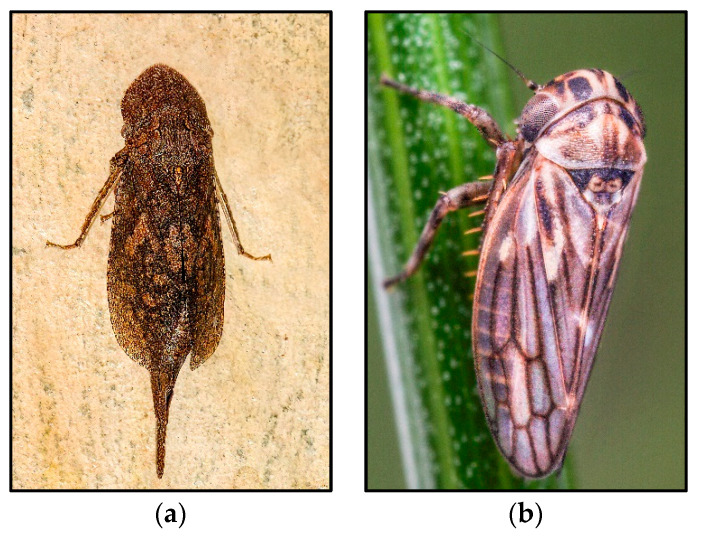
Examples from Cicadellidae in which parthenogenesis is suspected. (**a**) Ledrinae: *Ledromorpha planirostris*, photo credits: Dave Gray; (**b**) Megophthalminae: *Anaceratagallia frisia*, photo credits: Gernot Kunz.

**Table 1 insects-14-00820-t001:** Specialized terms used in this paper.

Term	Definition
Allopolyploidy	A form of polyploidy characterized by organisms possessing more than two sets of chromosomes originating from different species.
Apomixis	A reproductive process without fertilization in which meiosis and the fusion of gametes are partially or completely suppressed. Also known as ameiotic, apomictic parthenogenesis, or apomictic thelytoky.
Automixis	A form of reproduction without fertilization in which normal meiosis and the reduction of chromosome numbers occur. This is followed by the subsequent restoration of diploidy. Also known as meiotic, automictic parthenogenesis, or automictic thelytoky.
Arrhenotoky	A parthenogenetic mode of reproduction in which females exclusively produce male offspring from unfertilized eggs. Sometimes also referred to as androgenesis.
Autopolyploidy	A form of polyploidy in which organisms have more than two sets of chromosomes, all derived from the same parental species.
Contagious parthenogenesis	The reproductive process by which males, which periodically occur in a unisexual lineage, fertilize closely related “sexual” females. Subsequently, these fertilized females give rise to a new unisexual lineage through parthenogenesis.
Cyclical parthenogenesis	A life cycle with both sexual and asexual phases. This term is typically applied to cases in which the alternation between sexual and asexual reproduction is more or less regular and predictable.
Cytoplasmic incompatibility	A phenomenon in which the sperm and eggs of organisms are unable to form viable offspring due to modifications induced by intracellular parasites, such as *Wolbachia*, that reside within the cytoplasm of host cells.
Deuterotoky	A parthenogenetic mode of reproduction in which both male and female offspring are produced from unfertilized eggs.
Facultative parthenogenesis	A reproductive phenomenon in which sexual reproduction is the normal mode of reproduction for an organism; however, under certain conditions, a high percentage of the eggs have the capability to develop without fertilization. Also known as facultative thelytoky.
Haplodiploidy	A sex-determination system in which unfertilized haploid eggs develop into males while fertilized eggs develop into females.
Hermaphroditism	A condition in which an individual has both male and female reproductive organs.
Hybridization	The biological process or act of mating of individuals from different species resulting in the formation of hybrid offspring.
Geographical parthenogenesis	A phenomenon in which unisexual organisms, or parthenoforms, exhibit a different geographic distribution compared to their bisexual relatives. Parthenoforms tend to colonize regions characterized by higher latitudes, islands, and areas that were previously covered by glaciers.
Karyotype	The characterization of the chromosome complements of a species (such as the shape, type, number, etc.).
Obligatory parthenogenesis	A form of reproduction in which parthenogenesis is the sole mode of reproduction of an organism.
Parthenogenesis	A mode of reproduction characterized by the absence of fertilization (no fusion of sperm and egg nuclei). Also known as unisexual, uniparental, or asexual reproduction.
Paternal Genome Elimination (PGE)	A mode of reproduction in which males develop from fertilized eggs. However, during a certain stage of development, the complete haploid set of chromosomes inherited from their fathers is selectively eliminated and excluded from the sperm, resulting in males that are somatically haploid.
Paternal Genome Heterochromatinization (PGH)	A mode of reproduction characterized by the complete inactivation of the paternal genome through chromatin condensation. This process renders males functionally haploid, a condition known as parahaploidy.
Polyploidy	A genetic condition characterized by the presence of more than two sets of homologous chromosomes in the cells of an organism.
Pseudogamy	A reproductive phenomenon in which parthenogenetic development is initiated by the penetration of a sperm cell into an ovum. However, the sperm genome does not contribute to the genetic information of the resulting zygote. Also known as sperm-dependent parthenogenesis and gynogenesis.
Thelytoky	A form of reproduction in which female individuals produce exclusively female progeny by a process that does not involve fertilization. Thelytoky may be the sole mode of reproduction for a species or biotype (complete thelytoky), or it may alternate with sexual reproduction (i.e., cyclical thelytoky or heterogony) depending on environmental factors such as seasonal changes, temperature, or photoperiod.

**Table 2 insects-14-00820-t002:** Cases of true parthenogenesis in Auchenorrhyncha. Symbols: (!) need to be confirmed by lab-rearings; (*) some males reported; (Ø) mainly or only females known; (●) authors’ unpublished data; (_) no information; (?) species identification requires further confirmation; (A, B, C) morphotypes of *Empoasca*.

Species/Genera	Known Populations	Type of Parthenogenesis	Host Plants (Unisexuals/Bisexuals)	Chromosome Numbers (Unisexuals/Bisexuals)	Meiosis Type	Suggested Origin	Similarities to Bisexuals	Distribution of Bisexuals	Distribution of Unisexuals
*Delphacodes* cf *capnodes **	Bisexual !/Unisexual	Thelytokous	*Carex* *riparia/Carex* spp. *Eriophorum angustifolium*	3n = 44/ 2n = 30	Apomictic	_	Acoustic signals Morphology Mating behavior	Europe	Greece/ Germany !/Czeck Republic!
*Agallia* *quadripunctata*	Bisexual !/Unisexual	Thelytokous	*Trifolium* ! *Medicago* !, *Beta* !, *Ulmus* !, *Acer* !	Unknown/ Unknown	Unknown	_	Unknown	United States *, Cuba ! Mexico	Canada, Eastern United States Ø
*Empoasca* A	Unisexual	Thelytokous	Polyphagous	3n = 31 ●	Apomictic	Hybrid (?)	_	_	Madeira Island
*Empoasca* B	Unisexual	Thelytokous	Polyphagous	3n = 27 (?)	Apomictic	Autopolyploidy (?)	_	_	Madeira Island
*Empoasca* C	Unisexual/Bisexual !	Thelytokous	Polyphagous/*Ricinus*, *Phaseolus*, *Capsicum frutesceni*, *Gossypium*	3n = 24 ●	Apomictic	Autopolyploidy (?)	Morphology	Africa, Israel, Pakistan	Madeira Island
*Empoasca confusania* (?)	Unisexual/Bisexual !	Thelytokous	*Vigna unguiculata*, *Amaranthus*, Fabaceae spp./*Vigna unguiculata*, Fabaceae spp.	Unknown/ Unknown	Unknown	_	Morphology	Africa, Nigeria	Nigeria

**Table 3 insects-14-00820-t003:** Suspected cases of true parthenogenesis in Auchenorrhyncha. Symbols: *(*!) need to be confirmed by lab-rearings; *(**) some males reported; *(*Ø) mainly or only females known.

Family	Species	Type of Populations	Host Plants of Unisexuals	Distribution of Bisexuals	Distribution of Unisexuals
Issidae	*Scorlupella montana*	Bisexual !/ Unisexual !	*Carex* spp., *Eriophorum angustifolium*	United States *, Cuba ! and Mexico !	Greece/Germany !/Czeck Republic !
Cicadellidae	*Anaceratagallia kerzhneri*	Bisexual !/ Unisexual !	_	Eastern Russia	Mongolian steppe
	*Ledromorpha planirostris*	Unisexual !	*Eucalyptus* spp. Ø	Unknown	Eastern Australia from Queensland to Victoria and Tasmania Ø
	*Laburrus amazon*	Bisexual !/ Unisexual !	*Artemisia pauciflora*, *A. nitrosa* Ø	Lower Volga!	Southern European Russia, Kazakhstan, Mongolia Ø

## Data Availability

No new data were created or analyzed in this study. Data sharing is not applicable to this article.

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
