# Peer review of "True Parthenogenesis and Female-Biased Sex Ratios in Cicadomorpha and Fulgoromorpha (Hemiptera, Auchenorrhyncha)"

_insects, 2023, doi:10.3390/insects14100820_

Round 1
Reviewer 1 Report
The review by Pombo and Kuznetsova is aobut parthenogenesis in the Cicadomorpha and Fulgoromorpha. Although not a specialist in these insects, I noticed that the general terms are well used and that the topic was broadly reviewed. However, given that this article is a review mainly based in the cases of study made by the authors, I would expect some sort of future directions in these lines of research, some kind of prospective more than the two lines in the final conclusions. What else should be done to broaden this field of knowledge? I think a bit more should be said about this. OK, we know all that? On this basis, what more do we need to know? By stating a few lines, I think the paper would be publishable. In addition, some minor concerns should be address before publication.
Line 40: try to avoid the term “lower”. It is dismissed in Evolutionary Biology (see the T. Ryan Gregory paper published in Evo Edu Outreach (2008) 1:121–137)
Line 190: replace “reproduction biology” by “reproductive biology”.
Lines 190-194: CI is the only reproductive strategy induced by Wolbachia which do not skew the sex ratio towards females. This produces a sex ratio 1:1, but increases the amount of infected females at the expenses of uninfected females.
Species names should be given wherever they are alluded to for the first time along the text. Also, when named for the first time, give the name in full, even when the genus was already named. After that, the genus name should be abbreviated.
Line 378: replace “determined” by “diagnosed”, for example.
No additional comments
Author Response
Dear Reviewer, 1
Thank you for your valuable feedback on our work. We wholeheartedly agree with your comments, as they undoubtedly contribute to enhancing the overall interest of our research.
The review by Pombo and Kuznetsova is aobut parthenogenesis in the Cicadomorpha and Fulgoromorpha. Although not a specialist in these insects, I noticed that the general terms are well used and that the topic was broadly reviewed. However, given that this article is a review mainly based in the cases of study made by the authors, I would expect some sort of future directions in these lines of research, some kind of prospective more than the two lines in the final conclusions. What else should be done to broaden this field of knowledge? I think a bit more should be said about this. OK, we know all that? On this basis, what more do we need to know? By stating a few lines, I think the paper would be publishable. In addition, some minor concerns should be address before publication.
In light of this comment, we have incorporated the following paragraph, which we believe directly addresses the concerns raised:
As for the leafhopper genus Empoasca, to which our interests are addressed and our studies are devoted, new approaches are urgently necessary, first of all, studies at molecular-genetic level. Detection of several cases of parthenogenesis in one genus at once, while parthenogenesis in Auchenorrhyncha in general is extremely rare, testifies, apparently, to the lability of the sexual reproductive system in these leafhoppers. This also suggests that the genetic basis for the transition to parthenogenesis in Empoasca species likely involves a small number of genes. Therefore, further studies may be directed at identifying individual genes responsible for this reproductive transition and determining the mechanisms by which these genes cause such transitions. Our studies of the genus Empoasca have shown that Rickettsia bacteria are exclusively present in unisexual populations suggesting a connection between infection and parthenogenetic reproduction. Further research is needed, including searching and studying other parthenoforms of Empoasca, to prove such a relationship. Finally, it is important to understand whether parthenogenetic populations of Empoasca on Madeira confirm a prediction of the hypothesis of geographic pratenogenesis that unisexual reproduction tends to occur on islands.
Line 40: try to avoid the term “lower”. It is dismissed in Evolutionary Biology (see the T. Ryan Gregory paper published in Evo Edu Outreach (2008) 1:121–137)
We sincerely appreciate your comment. We completely agree with your perspective on the use of the term "lower" in evolutionary terms. Consequently, we have made the decision to remove it.
Line 190: replace “reproduction biology” by “reproductive biology”.
It was corrected as suggested.
Lines 190-194: CI is the only reproductive strategy induced by Wolbachia which do not skew the sex ratio towards females. This produces a sex ratio 1:1, but increases the amount of infected females at the expenses of uninfected females.
The affirmation is indeed accurate. In the text, we simply enumerate the various effects of microbial symbionts on the reproductive biology of its host, without presenting any opposing viewpoints. As a result, we have chosen to leave the text unchanged.
Species names should be given wherever they are alluded to for the first time along the text. Also, when named for the first time, give the name in full, even when the genus was already named. After that, the genus name should be abbreviated.
We ensure that all changes are made accordingly.
Line 378: replace “determined” by “diagnosed”, for example.
We make the change accordingly.
Thank you again for your input and for helping us improve the quality of our work.
Sinceramente
D. Aguín-Pombo e V. Kuznetsova
Reviewer 2 Report
This is a comprehensive and valuable review of the subject. I have marked one or two typos (typing errors) on the attached copy of the manuscript.

Author Response
Dear Reviewer, 2,
Thank you for your valuable feedback on our work. We sincerely appreciate your comments as they undoubtedly contribute to enhancing the overall interest of our research. We have taken your suggestions into consideration and made the necessary changes to the text accordingly.
Thank you again for your input and for helping us improve the quality of our work.
Sincerely,
D. Aguín-Pombo and V. Kuznetsova